# Paramedic Willingness to Report Violence Following the Introduction of a Novel, Point-of-Event Reporting Process in a Single Canadian Paramedic Service

**DOI:** 10.3390/ijerph21030363

**Published:** 2024-03-19

**Authors:** Justin Mausz, Michael-Jon Braaksma, Mandy Johnston, Alan M. Batt, Elizabeth A. Donnelly

**Affiliations:** 1Peel Regional Paramedic Services, 1600 Bovaird Drive East, Brampton, ON L6V 4R5, Canada; michael-jon.braaksma@peelregion.ca (M.-J.B.); mandy.johnston@peelregion.ca (M.J.); 2Faculty of Health Sciences, Queen’s University, 99 University Avenue, Kingston, ON K7L 3N6, Canada; alan.batt@queensu.ca; 3School of Social Work, University of Windsor, 167 Ferry Street, Room 213, Windsor, ON N9A 0C5, Canada; donnelly@uwindsor.ca

**Keywords:** paramedics, emergency medical services, emergency medical technicians, violence, workplace violence, qualitative research, survey research, mixed methods research

## Abstract

Violence against paramedics is increasingly recognized as an important occupational health problem, but pervasive and institutionalized underreporting hinders efforts at risk mitigation. Earlier research has shown that the organizational culture within paramedicine may contribute to underreporting, and researchers have recommended involving paramedics in the development of violence prevention policies, including reporting systems. Eighteen months after the launch of a new violence reporting system in Peel Region, Ontario, Canada, we surveyed paramedics about their experiences reporting violent encounters. Our objectives were to assess their willingness to report violence and explore factors that influence their decisions to file a report. Between September and December 2022, a total of 204 (33% of eligible) paramedics chose to participate, of whom 67% (N = 137) had experienced violence since the launch of the new reporting process, with 83% (N = 114) reporting the incidents at least some of the time. After thematically analyzing free-text survey responses, we found that the participants cited the accessibility of the new reporting process and the desire to promote accountability among perpetrators while contributing to a safer workplace as motivating factors. Their decisions to file a report, however, could be influenced by the perceived ‘volitionality’ and severity of the violent encounters, particularly in the context of (un)supportive co-workers and supervisors. Ultimately, the participants’ belief that the report would lead to meaningful change within the service was a key driver of reporting behavior.

## 1. Introduction

### 1.1. Violence against Paramedics

In a recent study reviewing ten years of data from the United States (US) Bureau of Labor Statistics, Emergency Medical Services (EMS) personnel were found to experience a risk of occupational injury from violence six times higher than the US population and 60% greater than comparable health professionals, such as nurses [1]. On average, the Bureau of Labor Statistics recorded 426 violence-related injuries each year, injuries that required either in-hospital medical treatment or resulted in lost time from work [1]. The investigation parallels a growing body of research internationally that characterizes violence against paramedics as a ‘serious public health problem’ [2] with the potential for significant physical [1,3,4,5,6,7,8,9,10,11,12,13,14,15] and psychological [16,17,18,19,20] harm. At the same time, recent years have seen numerous media reports of paramedics being seriously injured or killed after violent attacks by patients or members of the public [21,22,23,24,25]—a problem likely worsened amid eroding trust in institutions during the COVID-19 pandemic [26,27,28]. For example, in Canada, the federal government introduced changes to the Criminal Code to prohibit intimidating healthcare professionals in the course of their duties amid a rise in violent gatherings outside hospitals by people protesting public health measures [29,30,31]. Such protests included attacks on ambulances [32].

As a profession, paramedics in Canada and internationally experience high rates of work-related mental illness, including post-traumatic stress disorder (PTSD), major depressive disorder, generalized anxiety disorder, burnout, and suicide [33,34,35]. Extant research suggests that situations that involve threats to physical safety among paramedics increase the risk of psychological sequelae, including PTSD [33,36,37]. There is, then, a compelling reason to suggest that incidental and recurrent exposure to workplace violence creates an additional layer of risk, potentially compounding already high rates of exposure to occupational trauma [33] and contributing to adverse mental health outcomes for paramedics.

Mitigating the risk from violence to both physical and psychological health and safety requires evidence-informed violence prevention policy and supportive post-incident interventions for affected paramedics; however, there is now widespread recognition that—in all but the most egregious cases—individual incidents of violence often go unreported [2,7,10,38].

#### 1.1.1. A ‘Vastly Underreported’ Problem

Recent investigations into violence against healthcare workers estimated that at least 60% of incidents of violence against physicians, nurses, and allied health providers go unreported [27,39]—a problem that extends to paramedicine as well. In a 2014 survey of paramedics from two Canadian provinces, Bigham and colleagues found that more than 75% of paramedics had been exposed to some form of violence—including verbal abuse, threats, sexual harassment, or physical or sexual assault—within the past year [38]. However, despite negatively affecting the participants’ job satisfaction and home life, only 40% indicated that they had reported the incidents to supervisors or police and less than 20% documented the encounters through formal incident reporting mechanisms [38]. Their findings have since been replicated in other studies that increasingly recognize violence against paramedics as a ‘serious public health problem’ but one that remains ‘vastly underreported’ [13].

#### 1.1.2. The Role of Organizational Culture

While there may be administrative barriers to reporting created by inaccessible or cumbersome reporting processes, the organizational culture within paramedicine has been suggested as one potential factor that may limit reporting [38]. In 2019, we surveyed paramedics from Peel Region, Ontario, Canada, to inquire about their exposure to violence, whether they reported the incidents, and, if the participants chose not to file reports, what drove that underreporting behavior. In our study, despite nearly every participant indicating that they had been subjected to violence during their careers, only 40% reported the incidents to the service administration and just 20% reported the incidents to police [40]. In a qualitative analysis of the free-text survey comments provided by the participants, we identified a cyclical framework wherein chronic and widespread exposure to violence became perceived as unpreventable and without consequence for perpetrators [40]. Within this construction, the ability for paramedics to ‘brush off’ and ‘move on’ from violent encounters became normalized as an expected professional competency [40]—in essence, a learned behavior necessary to survive in the profession.

#### 1.1.3. The External Violence Incident Report

Reliable event-level data are key to any meaningful effort at violence risk mitigation, but solving for violence means first disrupting an existing organizational culture that positions violence as ‘just part of the job’ and ‘not worth’ reporting [40]. In their 2014 study, Bigham and colleagues [40] addressed this specifically in arguing that the existing reporting processes themselves may be an influential lever for promoting institutional change:
“It may be helpful for incident report forms to be designed specifically for reporting exposures to violence, and reporting may be more accepted if paramedics are involved with designing the reporting process”(p. 493)

Consequently, following our 2019 survey of paramedics in Peel Region, our team began developing a new reporting process to capture detailed information about violent encounters. The development and implementation of the External Violence Incident Report (EVIR) was described in an earlier publication [41]; but, in brief, the build involved extensive stakeholder consultation and pilot testing that, all told, lasted nearly a year. Throughout, the experience of the paramedic was centered in developing a user-friendly reporting mechanism that prioritized accessibility alongside comprehensive data collection. The result was a novel, point-of-event reporting process embedded within the electronic Patient Care Record (ePCR) that prompts paramedics to complete a violence report when filing an ePCR if they experienced violence during the call.

Apart from the form itself, the development included establishing a robust workplace violence prevention policy that outlined specific roles and responsibilities for both paramedics and the service administration. On receipt of a report, supervisors are accountable for attending to the immediate safety and psychosocial needs of the affected paramedics and (where appropriate) completing worker’s compensation forms, advocating for criminal charges to be brought against the perpetrator and flagging the perpetrator’s address if recurrent violent behavior is anticipated. Data from the reports are fed into a real-time situation awareness dashboard to identify trends that are then brought to an interoperability working group with the regional police service. Both the EVIR and associated ‘actions on’ process maps are available as Appendix A.

As a novel information source, data from the EVIRs are being used for research purposes, the goals of which have been explained elsewhere [42]. However, the purpose of this study was to reexamine the organizational culture that the EVIR sought to disrupt. Therefore, following eighteen months since the introduction of the new reporting process, our objectives were to reassess our paramedics’ willingness to report violence and explore factors that influenced their decisions on whether or not to report violent encounters.

## 2. Methods

### 2.1. Study Design

This study is the capstone project on a broader research program, with a detailed description provided in an earlier publication [42]. For this study specifically, we distributed a web-based cross-sectional survey to paramedics in Peel Region attending compulsory Continuing Medical Education (CME) sessions during the fall of 2022. The survey contained a mix of multiple-choice and free-text questions intended to explore their experiences with the new reporting process and identify factors that influenced their decisions about whether or not to report violent encounters. Paramedics choosing to participate were given protected time during their CME session to complete the survey.

### 2.2. Setting and Context

Our research is situated in the Region of Peel, in Ontario, Canada. Peel Regional Paramedic Services provides publicly funded land ambulance and paramedic services to the municipalities of Mississauga, Brampton, and Caledon, collectively encompassing approximately 1.5 million residents across a mixed suburban and rural geography of 1200 km^2^. At the time of the study, PRPS employed 750 Primary and Advanced Care Paramedics (P/ACPs) and approximately 60 paramedics in various administrative or supervisory roles. Collectively, PRPS personnel respond to an average of 130,000 emergency calls per year, making the paramedic service the second largest in the province by staffing and caseload.

The introduction of the new violence reporting process occurred alongside a suite of workplace violence prevention strategies launched following our initial study on barriers to reporting. These initiatives included new patient restraint equipment, comprehensive violence prevention policies, and a public-facing service position statement of ‘zero tolerance’ for violence against paramedics. Use of the new reporting process was encouraged by service supervisors and the workplace violence prevention program lead.

### 2.3. Data Collection

Paramedics in Peel Region are required to attend in-person CME sessions held in the fall of each year. Sessions are organized by the paramedic service and paramedics are required to attend on a scheduled day off but are paid their usual wage for the day. We invited paramedics attending the fall 2022 CME to participate in a web-based survey as part of our program evaluation strategy following the introduction of the new reporting process. All paramedics attending CME were eligible to participate and we had no criteria to exclude potential participants.

The survey was composed of four sections: the first inquired about the paramedics’ experiences with violence and their use of the new reporting process since its introduction in February 2021. This section primarily used multiple-choice response formats for questions about whether the paramedic had been subjected to violence, how often they reported the incidents, and whether their experiences with the new reporting process encouraged them to report similar incidents in the future. The second section focused more specifically on factors that influenced the paramedics’ decisions to report violent encounters. The paramedics were asked a series of open-ended questions about why they did or did not report a particular incident, and—if they chose not to report the incident—what the service administration could do to encourage them to report incidents in the future. These questions required the paramedics to provide a typed response but imposed no word or character limit on their answers. The third section asked paramedics to rate their perceptions of the utility of several violence prevention initiatives implemented within the paramedic service on a 5-point Likert scale. These questions were very specific to the paramedic service and were intended for program evaluation purposes; consequently, the data are not reported here. Finally, the survey concluded with demographic questions that inquired about the participant’s gender, employment classification (e.g., part- or full-time), clinical certification level (PCP or ACP), role within the service (e.g., front line vs. leadership/supervisory role), and years of experience within paramedicine.

The survey was housed on Microsoft Forms with a QR code made available for paramedics to access the survey during the CME session. Consenting paramedics were given a Wi-Fi-enabled tablet to complete the survey electronically during a break in the curriculum content. Pilot testing suggested the survey would take an average of about 10 min to complete. The full text of the survey is included as Appendix A.

### 2.4. Data Analysis

We analyzed the data from the survey in two ways. First, we used frequency counts to report on the responses to multiple-choice questions and explored group differences across demographic categories using chi-squared tests. Here, our outcome of interest was reporting behavior in response to the question ‘did you report the incident(s)?’ We assessed this by collapsing the options ‘yes’ and ‘sometimes’ into an ‘reported at least sometimes’ variable compared to ‘no’ responses. All analyses were carried out in SPSS Statistics (IBM Corporation; Version 28) and we followed convention in accepting a *p*-value of <0.05 with confidence intervals that exclude the null value as indicating statistical significance.

For the qualitative data, we aggregated the free-text narrative responses by question and imported the data into NVivo for Mac (QSR International; Version 11) for thematic analysis. In scrutinizing the free-text responses, we followed the principles of qualitative content analysis as explained by Vaismoradi and colleagues [43]. This involved subjecting the data to several rounds of open [44] followed by focused [45] coding with the goal of identifying themes but while remaining analytically ‘close’ to the original text. Because the qualitative analysis of survey comments is subject to well-recognized limitations [46], we were cautious not to draw too deeply in our analytical inferences. Sensitized by our earlier work on potential barriers to reporting, we were interested specifically in understanding factors that could sway a paramedic to being more or less likely to report a violent encounter at work.

## 3. Results

### 3.1. Response Rate and Participant Characteristics

Between 1 September 2022 and 31 January 2023, a total of 619 active-duty (i.e., not on leave) paramedics attended CME, of whom 204 (33%) agreed to participate.

Among participants, 54% (N = 110) were men, 40% (N = 86) were women, and a small number (<10) either declined to disclose their gender or provided another, non-binary gender option. The majority (70%; N = 143) were certified at the primary care level, and most (59%; N = 120) were employed full-time. The participants were roughly evenly split between early- (32%; N = 66) or mid-career (36%; N = 74), and most (93%) were in a front-line service delivery role, with 12 participants (6%) currently in a supervisory or leadership position. Demographic details are summarized in Table 1.

### 3.2. Exposure to Violence and Willingness to Report

Among participants, 137 (67%) indicated they had experienced violence since the launch of the new reporting process, with most (47%; N = 96) experiencing between one and five incidents, and some (14%; N = 29) experiencing between 6 and 10 incidents. Twelve participants (~6%) indicated having experienced more than 10 incidents of violence since the launch of the new reporting process.

Among those experiencing violence, 52% (N = 72) indicated they always reported the incidents, with an additional 30% (N = 42) indicating they sometimes reported the incidents using the new reporting process; taken together, 83% indicated they reported the incidents at least some of the time. Within the group reporting violence at least some of the time, 86% (N = 98) indicated that their experience with the reporting process) encouraged them to report similar incidents in the future. Conversely, 14% (N = 16) indicated their experiences discouraged them from reporting similar incidents in the future (Figure 1).

In assessing group differences in willingness to report, neither gender, career stage, nor clinical certification were associated with reporting behavior (*p* = 0.09, 0.22, and 0.09, respectively). We did observe that those employed full-time were more likely to report violent incidents (Odds Ratio [OR] 2.78, 95% CI 1.11–6.93, *p* = 0.025).

### 3.3. Thematic Analysis Overview

In qualitatively analyzing the free-text survey responses, we identified three key motivators that enabled reporting: the accessibility of the reporting process, the desire to hold perpetrators to account, and a felt sense of responsibility in contributing positively to workplace violence prevention. We also identified modifiers that could ‘sway’ a paramedic’s decision toward being more or less likely to report; specifically, the perceived ‘volitionality’ and severity of a particular incident, co-worker (including supervisor) support, and the anticipated utility of the report in preventing violence in the future. Each is discussed in detail below.

### 3.4. Motivators and Enablers of Reporting

The participants explained that the accessibility of the new reporting process was a powerful motivator for filing reports. In our development phase, we took great care to ensure that the administrative burden on the paramedics would be minimized to the greatest extent reasonably possible. It was encouraging, then, to see that the survey participants were citing accessibility as an enabling factor in reporting:
“The form was short and easy to fill out, and I like the idea that my report can make a tangible difference for the experience of other paramedics”. “It was simple and straightforward”.

Interestingly, the accessibility of the reporting process appeared to have an additional—and admittedly unanticipated—benefit. In providing an accessible and user-friendly platform to report violent encounters, the free-text narrative description of the report form gave the paramedics a ‘container’ in which to offload potentially distressing details of their experiences:
“I just felt justified and validated in filing a report and (it) normalized (me) feeling angry at what happened. It was a healthy way to share my story, even if it was just on paper”.

In keeping with the paramedic service’s public position of zero tolerance for violence against its staff, the participants cited accountability for perpetrators as a key driver for reporting. For example, one participant said “I (wanted) to show that it’s not okay”, while another said they reported because they were trying to “incarcerate the perpetrator”. At the same time, the participants also explained that reporting violent encounters is an important strategy to promote co-worker safety in the event paramedics attend a particular residence or patient again in the future:
“Wanting the data to help promote action to address violence against paramedics and have hazard flags (i.e., on the patient’s address) to mitigate violence against other responders”.
“I truly believe this program makes a difference. Getting more data (from reports) will help you solidify the program”.

### 3.5. Potentially Influential Modifiers

Notwithstanding the accessibility of the reporting process, we identified that the participants’ perceptions of the ‘volitionality’ and severity of the violence appeared to have significant influence on their decisions to file a report. Incidents that were felt to be less serious or not intended to cause harm were less likely to be reported:
“Most (of the incidents) were people with legitimate medical issues”. “Verbal abuse I do not always report”. “The patient did not intend to be violent, (their behavior) was caused by (a) mental health issue”. “It was just verbal abuse from impaired individuals”.

The influence of co-workers—including other paramedics and supervisors—could tip the scales on whether a paramedic was inclined to file a violence report. For example, in some cases, (usually junior) paramedics were reluctant to file a report because they thought it would portray the more senior paramedic in an unfavorable light for not managing the 9-1-1 call differently:
“Sometimes I was forced not to (file a report) because my partner was against it or they thought the report would reflect badly on them”.

Conversely, support from other paramedics or from supervisors who were involved in the 9-1-1 call could encourage a participant to file a report:
“Support from coworkers and supervisors”. “Supervisor recommendation on scene”. “Supervisor support. Knowing (the report) will be flagged and lower the risk of a different paramedic experiencing violence with this individual”.

Critically, however, the participants explained that their decisions to file a report hinged on their belief that the report would be taken seriously and lead to meaningful change. For example, the participants were disinclined to file future reports when they received an apathetic response to a report they submitted, or no response at all: “I am disappointed there’s been no follow up on any of my reports”. On the other hand, experiences in which the reports were actioned promptly were powerful motivators for encouraging future reporting:
“I understand the value of reporting to support change within the profession. Even if not impacted myself, I (file reports) to keep my co-workers safe. I see the efforts of the service to address violence and appreciate them. Reporting is me doing my part”.

## 4. Discussion

Following the launch of a novel reporting process in February 2021, our objective was to reassess our paramedics’ willingness to report violent incidents and explore factors that influenced their decisions to file reports after 18–24 months with the new system. With a response rate and sample that are comparable to our initial study on barriers to reporting in our service [40], we found that paramedic willingness to report has more than doubled after introducing the new reporting process and associated violence prevention programming.

Our survey participants explained that the accessibility of the reporting process combined with a desire to hold perpetrators accountable and contribute positively to workplace violence prevention were powerful motivators that drove their increased willingness to report. Importantly, however, the participants’ perceptions of the ‘volitionality’ and severity of the violence, combined with the support (or lack of support) from coworkers and supervisors, could influence a paramedic’s decision to file a report—particularly if they felt that the report may not be actioned promptly. In that respect, our findings underscore the role of organizational culture in either contributing to or threatening paramedics’ willingness to report violent encounters with the public. Our work has important implications for both research and policy in this space.

First, and from a research perspective, several investigations into violence against healthcare workers broadly [28,47,48] and paramedics in particular [2,10,13,14,38] have underscored that healthcare workers and paramedics alike consider violence ‘just part of the job’. What has been less clear, however, is how such a potentially serious threat to health, safety, and well-being can be so casually dismissed in such sentiments. Where both our earlier [40] and current work contribute is in providing additional granularity on the underlying mechanisms of organizational culture that sustain underreporting. In our 2019 study, we saw that widespread and chronic violence becomes perceived as unpreventable and without consequence for perpetrators. Our efforts since then are aimed at disrupting that cycle, but what we saw in the current work were two important dimensions that influenced reporting behavior: the perceived severity and ‘volitionality’ of the violence; and various factors that led a paramedic to believe that an individual violence report will lead to meaningful change in the future risk of violence. To a large degree, volitionality and severity are in the eye of the beholder; what may be serious and intentionally hurtful to one paramedic may not be to another. The tapestry of individual perspectives on both points makes up the organizational culture within which change may be possible. In that respect, peer and supervisor support was a large factor in willingness to report because it influenced the paramedics’ beliefs that their reports would be taken seriously and put to meaningful use. This framework is useful because it begins to prescribe a path forward for policy. Finally, our findings also serve as a useful case-in-point for Bigham and colleagues’ earlier suggestion that reporting may be more accepted if paramedics are involved in developing purpose-built incident reporting processes [38].

From a policy perspective, our findings illustrate that changing reporting behavior is indeed possible, if not particularly easy. In choosing to file a report, our participants often cited encouragement from peers and—particularly—supervisors in assuring them that their reports would be put to meaningful use. This suggests that in creating workplace violence prevention policies, the role of supervisors cannot be overstated. When supervisors actioned the reports promptly and provided feedback to the paramedics, the participants said this encouraged them to report similar incidents in the future. With sufficient stakeholder ‘buy-in’, this process can become self-sustaining. Importantly, however, we should note that the work to get to this point was not easy. The development of violence reporting mechanisms and the associated follow-up processes has been both a costly and resource-intensive undertaking. While the psychosocial benefits of a more supportive organizational culture are important in their own right, we are still far from being able to say that the work is driving a significant and sustained decrease in *exposure* to violence. On that front, the paramedic service has just recently been approved for comprehensive violence avoidance training—a program that will cost approximately CAD 1.25 million [49].

### Limitations

Our findings should be interpreted within the context of certain limitations. First, we acknowledge that our response rate of 33% is admittedly low. The timing of data collection meant that our survey was distributed alongside a backlog of other routine administrative and program evaluation surveys that had been deferred when earlier in-person CME sessions had been canceled for public health reasons during the COVID-19 pandemic. Combined with staffing shortages and growing COVID-19 case counts, the workplace climate during data collection was underscored by fatigue. Second, we are cognizant that the qualitative analysis of free-text survey comments does not yield the same depth of insights that are achievable through in-depth interviews or focus groups, for example. We structured our questions to encourage broad responses on specific points of interest in an attempt to strike a balance between the feasibility advantages of the survey approach with its limitations. Third, as with much qualitative inquiry, our findings are contextually situated and readers should take care in extrapolating or applying our results in their own settings. Finally, as is unfortunately the case with many surveys, our findings are vulnerable to a degree of selection bias—perhaps more so given the strained work environment within which the study was carried out.

## 5. Conclusions

With a response rate and sample that are comparable to our earlier study, we found that paramedic willingness to report violent encounters has more than doubled in our service following the introduction of a new reporting process alongside a broader violence prevention program. Willingness to report was generally consistent across demographic groups; however, we did observe that full-time employees appeared more willing to report incidents for reasons that are not immediately clear. The accessibility of the reporting process, combined with the perceived volitionality and severity of violent encounters and the influence of coworkers were key determinants of whether paramedics elected to submit a report. Ultimately, the paramedics’ belief that the report would lead to meaningful change was a key driver of reporting behavior. Our findings add further evidence to the role of organizational culture in either promoting or stigmatizing reporting behavior and demonstrate that, with organizational commitment, norms around reporting violence can indeed change.

## Figures and Tables

**Figure 1 ijerph-21-00363-f001:**
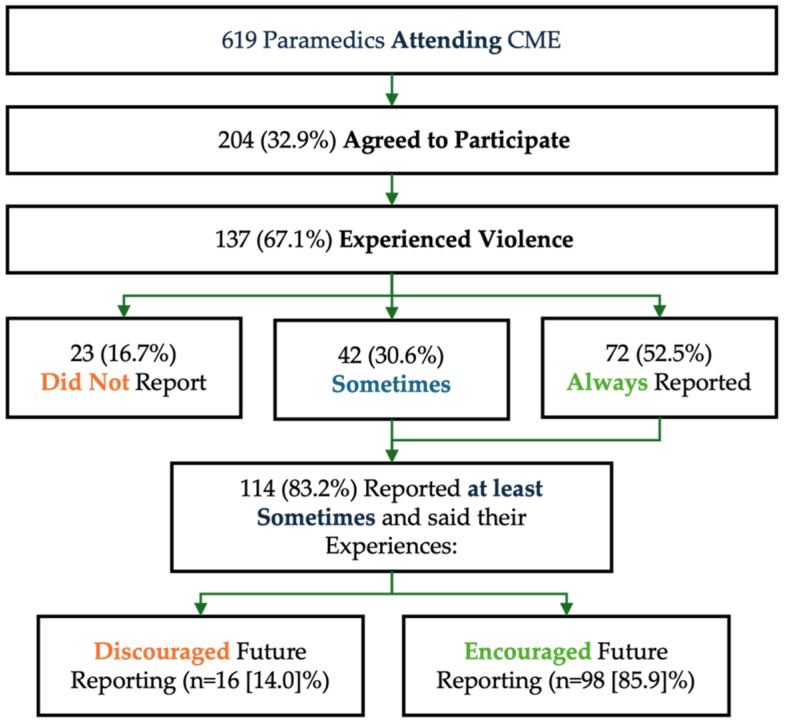
Participant flow overview. Note that green text is used to indicate more favorable responses.

**Table 1 ijerph-21-00363-t001:** Experience with violence and reporting behavior stratified by demographic characteristics.

Parameter	N	%	Experienced Violence (N [%])	Reported at Least Sometimes (N [%])	Did Not Report (N [%])	*p*-Value
Gender						0.093
Men	110	53.9%	72 (65.4%)	58 (80.5%)	14 (19.4%)	
Women	86	42.1%	61 (70.9%)	54 (88.5%)	7 (11.4%)	
Other *	8	3.9%	4 (50%)	2 (50%)	2 (50%)	
Clinical Certification						0.097
Primary Care	143	70.4%	93 (65%)	74 (79.5%)	19 (20.4%)	
Advanced Care	60	20.5%	44 (73.3%)	40 (90.9%)	4 (9%)	
Missing	1					
Experience						0.220
New (<1 year)	18	8.8%	9 (50%)	6 (66.6%)	3 (33.3%)	
Early Career (1–4 years)	66	32.5%	47 (71.2%)	37 (78.7%)	10 (21.2%)	
Mid-Career (5–15 years)	74	36.4%	53 (71.6%)	45 (84.9%)	8 (15%)	
Senior (>15 years)	45	22.1%	28 (62.2%)	26 (92.8%)	2 (7.1%)	
Missing	1					
Employment						0.025
Part-Time	82	40.5%	49 (59.7%)	36 (73.4%)	13 (26.5)	
Full-Time	120	59.4%	87 (72.5%)	77 (88.5%)	10 (11.4%)	
Missing	2					

Note: * Other category includes participants who declined to disclose their gender or who provided another, non-binary gender. Reporting percentages refer to participants who experienced violence. *p*-values refer to chi-squared tests between any vs. no reporting for each stratum. Data collected via web-based survey in Brampton Ontario from September to December 2022. Overall N = 204.

## Data Availability

Data for this study may be shared with interested researchers on a case-by-case basis, subject to a privacy review and formal data sharing agreement.

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
