# Peer review of "Paramedic Willingness to Report Violence Following the Introduction of a Novel, Point-of-Event Reporting Process in a Single Canadian Paramedic Service"

_ijerph, 2024, doi:10.3390/ijerph21030363_

Round 1
Reviewer 1 Report
Comments and Suggestions for Authors
The study is relevant. I list these considerations to guide the writing of this study:
ABSTRACT: Include the objective of the study in the abstract. Inform the study design. Include the year in which the data was collected. Include the variables that were analyzed. Include the data analysis method.
KEYWORDS: check the limit of keywords allowed by the journal.
INTRODUCTION: I suggest listing some characteristics of the victim that are statistically associated with higher incidences of violence.
INTRODUCTION: The authors present the results of their study in the introduction (1.1.2). I suggest presenting the results and discussion after explaining the study method.
METHODS: describe the study design.
METHODS: Were there any criteria for excluding participants?
METHODS: Describe the variables and open questions used. The way the 4 sections of the research are described, it is not clear how the questions were asked or which variables will emerge. - Make clear the group of variables that were used in the quantitative analysis, and make clear the open questions that were used to apply the qualitative analysis.
METHODS: For the quantitative part of the study, I suggest reporting the outcome and the independent variables.
TABLE 1: Table 1 is edited in "table" format. Adjust formatting to table.
TABLE 1: Complete the information in the title of the table, include the city where the data was collected, the year and the number of participants analyzed.
FIGURE 1: Check what should be rounded, the sum of these values should be 100%: ["17% Did not report"] + ["30% sometimes"] + ["52% always reported"].
DISCUSSION: the discussion part needs to develop a greater dialog between the results of this study and other publications that have carried out similar studies.
REFERENCES: try to update references published more than 5 years ago. At least 40% of the references have been published for more than 5 years.
Author Response
Dear colleague,
Thank you for taking the time to review our manuscript and for providing helpful feedback. We have incorporated many of the edits you suggested and believe this has strengthened our work as a result. Please refer to the table below for a more detailed response to each of the points you raised.
Thank you kindly,
Justin
|
Comment |
Response |
Page/Line # |
|
ABSTRACT: Include the objective of the study in the abstract. Inform the study design. Include the year in which the data was collected. Include the variables that were analyzed. Include the data analysis method. |
We have reformatted the abstract per your suggestions, with the exception of the variable point. Robust hypothesis testing was not a primary objective of this study. |
Page 1, Lines 17-24 |
|
KEYWORDS: check the limit of keywords allowed by the journal. |
The journal asks for 3 to 10 keywords; we have 8. |
N/A |
|
INTRODUCTION: I suggest listing some characteristics of the victim that are statistically associated with higher incidences of violence. |
Unfortunately, we do not have data on this, however, it is the subject of a future study currently in the planning stages. |
N/A |
|
INTRODUCTION: The authors present the results of their study in the introduction (1.1.2). I suggest presenting the results and discussion after explaining the study method. |
Actually, these results are from a study we did in 2019. The results are attributed to a citation, and we included this earlier work to scaffold the rationale for our current study. |
N/A |
|
METHODS: describe the study design. |
Paragraph 2.1 relabelled to “Study Design” and ‘cross-sectional’ added to description of survey |
Page 3, Paragraph 2.1 |
|
METHODS: Were there any criteria for excluding participants? |
No |
N/A |
|
METHODS: Describe the variables and open questions used. The way the 4 sections of the research are described, it is not clear how the questions were asked or which variables will emerge. - Make clear the group of variables that were used in the quantitative analysis, and make clear the open questions that were used to apply the qualitative analysis. |
The full text of the survey, including wording of questions and logic flow of the prompts is provided in Figure S3. |
N/A |
|
METHODS: For the quantitative part of the study, I suggest reporting the outcome and the independent variables. |
We have added relevant language in Methods paragraph 2.3 (Data Analysis) |
Page 4, Lines 191-194 |
|
TABLE 1: Table 1 is edited in "table" format. Adjust formatting to table. |
I’m afraid I’m not clear on what your comment here is referring to. |
N/A |
|
TABLE 1: Complete the information in the title of the table, include the city where the data was collected, the year and the number of participants analyzed. |
Reformatted per your suggestion |
Page 5, Table 1 |
|
FIGURE 1: Check what should be rounded, the sum of these values should be 100%: ["17% Did not report"] + ["30% sometimes"] + ["52% always reported"]. |
We are within acceptable parameters for rounding for IJERPH; the values here sum to 99%. We have added a note in the caption. |
Page 6, Figure 1 |
|
DISCUSSION: the discussion part needs to develop a greater dialog between the results of this study and other publications that have carried out similar studies. |
|
|
|
REFERENCES: try to update references published more than 5 years ago. At least 40% of the references have been published for more than 5 years. |
When I added them up, 71% of our references were published ≥2018. I think that’s reasonable – we’re not even 1 full quarter into 2024 and 2020-2022 would have understandably put a dent in non-pandemic related research. |
N/A |
Reviewer 2 Report
Comments and Suggestions for Authors
1. In the background, can the author add to the problem of why violence that occurs in the workplace is not reported? What factors influence this?
2. In the background, especially problems related to violence at the study location, can you mention the history of violence that has occurred?
3. On page 6, especially in section 3. Results, the number of paramedics who were willing to take part in the study was 204 participants. However, on page 7, in employment, the total number of participants is only 202 (82 Part-Time workers + 120 Full-Time workers). Please check the numbers and percentages in Table 1 more carefully!
4. In Table 1 page 6, in the parameters gender, Clinical Certification, Experience, and Employment, it is found that N% when added up does not show a total of 100%, please check again how the percentage calculation has been made
5. in the discussion, especially in the first and second paragraphs on pages 9-10, the author only writes opinions from the results of studies that have been carried out, it would be better if the authors could add analysis based on theory or related studies
6. When writing a manuscript, please check the suitability of objectives, results and conclusions. The objective states "explore factors that influence their decisions on whether or not to report violent encounters". In Table 1 there are 4 parameters related to factors that have been tested statistically, but the conclusion does not touch on any of the factors related to these parameters
7. on page 3 paragraphs 1 and 2, the author quotes too many of his reference sources (citation number 40) in 1 paragraph, please look for other relevant sources
8. What research methods were used in this study? Does this use a mixed-method approach? the author can write it in the Methods section
Comments on the Quality of English LanguageMinor editing of the English language required
Author Response
Dear colleague,
Thank you for taking the time to review our manuscript and for providing such helpful feedback. You raised several important points and we have taken care to incorporate many of your suggestions into our revised manuscript. Your comments have undoubtedly strengthened the quality of our work. Please refer to the below table for more detailed information about the edits.
Thank you kindly,
Justin
|
Comment |
Response |
Page/Line # |
|
In the background, can the author add to the problem of why violence that occurs in the workplace is not reported? What factors influence this? |
The reasons for underreporting are listed in 1.1.2 (The role of organizational culture), where we illustrate that violence is underreported because it is perceived as widespread, unpreventable, and without consequence to the perpetrators. |
Page 2, Lines 87-92 |
|
In the background, especially problems related to violence at the study location, can you mention the history of violence that has occurred? |
Prior to the development of the EVIR, we do not have robust data to point to the historical experiences of violence. We did include the only study that exists within our (Canadian) population points to 75% of paramedics experiencing violence (Bigham, 2014).. |
Page 2, Lines 68-70 |
|
On page 6, especially in section 3. Results, the number of paramedics who were willing to take part in the study was 204 participants. However, on page 7, in employment, the total number of participants is only 202 (82 Part-Time workers + 120 Full-Time workers). Please check the numbers and percentages in Table 1 more carefully! |
Are you referring to Table 1 on page 5? Two participants did not answer this question – that’s why we came up a bit short there. We added line to each row of the table to indicate the number of participants who did not answer a particular question. Fortunately, this is a very small number. |
Page 5, Table 1 |
|
In Table 1 page 6, in the parameters gender, Clinical Certification, Experience, and Employment, it is found that N% when added up does not show a total of 100%, please check again how the percentage calculation has been made |
Correct, just because we had the occasional instance where a participant did not answer a question. |
Page 5, Table 1 |
|
In the discussion, especially in the first and second paragraphs on pages 9-10, the author only writes opinions from the results of studies that have been carried out, it would be better if the authors could add analysis based on theory or related studies |
In the first two paragraphs, we are summarizing the findings from the current study and in the remaining paragraphs, we are contextualizing them within the relevant broader research literature. I can understand how this may appear as opining, but it is intended as a scholarly discussion of empirically generated findings. |
N/A |
|
When writing a manuscript, please check the suitability of objectives, results and conclusions. The objective states "explore factors that influence their decisions on whether or not to report violent encounters". In Table 1 there are 4 parameters related to factors that have been tested statistically, but the conclusion does not touch on any of the factors related to these parameters |
We have added language to the conclusion emphasizing that full-time (vs. part-time) employees appeared more inclined to report violent incidents. |
Page 11, Lines 381-383 |
|
on page 3 paragraphs 1 and 2, the author quotes too many of his reference sources (citation number 40) in 1 paragraph, please look for other relevant sources |
Citation #40 is actually quite foundational to the work. That study (done in 2019) identified barriers to reporting and we used the results from that study to design a new reporting process. In the current study, we are re-assessing willingness to report following ~18 months with the new reporting system. |
N/A |
|
What research methods were used in this study? Does this use a mixed-method approach? the author can write it in the Methods section |
We used a cross-sectional, web-based survey distributed to a cohort of paramedics in a single paramedic service in Ontario, Canada. We describe this in methods paragraph 2.1 (“Study Design”) |
Page 3, Methods, Paragraph 2.1s |
Reviewer 3 Report
Comments and Suggestions for Authors
This is a well-written paper describing the results of an initiative to encourage reporting of violent incidents by paramedics in a Canadian province. The authors describe the process of developing the reporting tool, which included significant stakeholder input and a strong desire for the reporting process to have minimal burden on the reporting employee. Impact of the reporting tool was ascertained by a survey, which included an opportunity to include free text answers.
Strengths
- The manuscript is well-written, with clarity and conciseness.
- The current work describes part of a long-standing initiative to address the problem of increasing violence directed at paramedics.
- The table and figure provide valuable information.
- The material in the appendices is very useful and well done.
- The paper demonstrates a significant change in attitudes and behaviors regarding reporting of violence.
- Paramedics appear empowered and it is hoped that sustained action results from this important initiative. (Downstream results of the reporting is beyond the scope of this manuscript.)
Weaknesses
- There are very few weaknesses.
- The survey response rate of 33%, cited as a weakness, is not surprising, given the multiple factors impinging on paramedics’ workload, time, and interest in completing surveys.
- There may be some self-selection bias from who elected to respond to the survey.
Author Response
Dear colleague,
Thank you for taking the time to review our manuscript and for offering such encouraging feedback. We did make one edit in our paper in response to the point you raised about the possibility of selection bias. We agree entirely – it is, unfortunately, a common failing of surveys, particularly those with low response rates, and even though you did not push the point strongly, we felt it worth adding to the limitations section, given that we offered our survey during a time of considerable strain in the workplace. You will see this reflected on page 9, paragraph 4.1, lines 376 through 378.
Again, thank you for reviewing our paper and for your constructive comments.
Justin
Round 2
Reviewer 1 Report
Comments and Suggestions for Authors
I congratulate the authors on their adjustments to the text. I note that they have paid attention to the considerations sent in.
I would like to highlight these points for you to evaluate:
- Check whether it is common and acceptable to write that "Note that some values may not sum to 100% due to rounding." [Line 242]. The conventional thing to do would be to round the values so that they add up to 100%. It's a small adjustment, so as not to generate unnecessary noise.
- METHODS: In topic 2.3 "2.3. Data Collection", write down whether there were any criteria for excluding participants, and give reasons if there were.
- The number of topic "2.3" is being repeated: 2.3. Data Collection ; 2.3. Data Analysis [PAGE 4]
- I note that the study's data analysis does not include a measure of association to analyze the comparison of the relationship between the independent variables and the dependent variable. This would be an interesting way of exploring the database available.
Author Response
Dear colleague,
Thank you again for providing another helpful and constructive review of our manuscript and for returning the paper in such a timely fashion. As was the case with your comments during the first round, your feedback has been helpful in strengthening the quality of our work and we have incorporated nearly all of your suggested edits. Please refer to the below table for more specific information and accept our gratitude for the time and effort you invested in this review.
Kindly,
Justin
